# In Silico Screening for Pesticide Candidates against the Desert Locust *Schistocerca gregaria*

**DOI:** 10.3390/life12030387

**Published:** 2022-03-07

**Authors:** Graham E. Jackson, Gerd Gäde, Heather G. Marco

**Affiliations:** 1Department of Chemistry, University of Cape Town, Cape Town 7701, South Africa; 2Department of Biological Sciences, University of Cape Town, Cape Town 7701, South Africa; gerd.gade@uct.ac.za (G.G.); heather.marco@uct.ac.za (H.G.M.)

**Keywords:** adipokinetic hormone, desert locust, *Schistocerca gregaria*, in silico screening, ZINC20

## Abstract

Adipokinetic hormone (AKH) is one of the most important metabolic neuropeptides in insects, with actions similar to glucagon in vertebrates. AKH regulates carbohydrate and fat metabolism by mobilizing trehalose and diacylglycerol into circulation from glycogen and triacylglycerol stores, respectively, in the fat body. The short peptide (8 to 10 amino acids long) exerts its function by binding to a rhodopsin-like G protein-coupled receptor located in the cell membrane of the fat body. The AKH receptor (AKHR) is, thus, a potential target for the development of novel specific (peptide) mimetics to control pest insects, such as locusts, which are feared for their prolific breeding, swarm-forming behavior and voracious appetite. Previously, we proposed a model of the interaction between the three endogenous AKHs of the desert locust, *Schistocerca gregaria*, and the cognate AKHR (Jackson et al., Peer J. 7, e7514, 2019). In the current study we have performed in silico screening of two databases (NCI Open 2012 library and Zinc20) to identify compounds which may fit the endogenous Schgr-AKH-II binding site on the AKHR of *S. gregaria*. In all, 354 compounds were found to fit the binding site with glide scores < −8. Using the glide scores and binding energies, 7 docked compounds were selected for molecular dynamic simulation in a phosphatidylcholine membrane. Of these 7 compounds, 4 had binding energies which would allow them to compete with Schgr-AKH-II for the receptor binding site and so are proposed as agonistic ligand candidates. One of the ligands, ZINC000257251537, was tested in a homospecific in vivo biological assay and found to have significant antagonistic activity.

## 1. Introduction

The generic name “locusts” describes a small number of insect species from the orthopteran suborder Caelifera (short-horned grasshoppers) that are characterized by a phenomenon called “phase polyphenism”, which is defined as a change in behavior and morphological appearance on account of the density of the population [1]. The desert locust, *Schistocerca gregaria*, is one of the best researched examples in this respect and is one of the most destructive pest insects known to mankind [1]. *S. gregaria* may occur in two phases: when population density is low, the insects are in the solitary phase; adults have cryptic coloration, they avoid one another and are nocturnally active. The gregarious phase occurs at high population density and is accompanied by changes in morphology, behavior and, also, physiology; individuals are active diurnally, they aggregate and in the larval stage form hopper bands that are constantly on the “march” devastating grasslands by voracious feeding; the adult insects form huge migratory swarms that conquer new habitats through flight, feeding on almost all available plant material [2,3]. Such swarms are no novelty and have been reported in the Book of Exodus (Old Testament) as one of the seven plagues in ancient Egypt (The Bible). In 2019/2020, huge swarms were detected in many countries north of the equator, resulting in the worst negative economic impact on regional farmers since the last 70 years.

The chemical answer to such pest outbreak problems is invariably spraying the hopper bands and the roosting adult swarms with insecticides, in total disregard of putative negative effects to other organisms (including humans) and to the environment. An alternative strategy is to use so-called “green” insecticides that are characterized by a biorational design to be specific to the target species, thus selective to harm the pest species but to have little or no effect on other species. In recent years, a consortium has extensively studied various insect neuropeptides in this context (www.neurostresspep.eu, accessed on 6 February 2022). One of the neuropeptides under study is the metabolic adipokinetic hormone (AKH) which had previously been proposed as a potential insecticidal compound [4]. The major function of an AKH in an insect is the control of metabolism, especially during intensive muscular activity, such as flight or swimming and, hence, it can be compared functionally with the vertebrate hormone glucagon [5,6,7]. Structurally, however, the mature AKH peptide and its G protein-coupled receptor (GPCR) is rather more related to the vertebrate gonadotropin releasing hormone (GnRH) signaling system; in fact, GnRH, AKH and two other insect neuropeptides—corazonin and the adipokinetic hormone/corazonin-related peptide—are suggested to form a large superfamily [8,9,10]. AKH is synthesized in and released from the neurohemal corpus cardiacum into circulation. With respect to the primary structure, the mature AKH peptides are characterized by a chain length of 8 to 10 amino acids and post-translationally modified amino—(pyroglutamate) and carboxy (amidation) termini. At position two from the amino terminal, AKHs have either an aliphatic amino acid (leucine, isoleucine or valine) or an aromatic amino acid (phenylalanine); position three is either a threonine or asparagine residue; at position four one finds either the aromatic phenylalanine or tyrosine; position five has either threonine or serine; position eight is always the aromatic tryptophan and position nine comprises of a simple glycine, whereas at position six, seven and ten there is a large variability of amino acids possible [11]. Currently, more than 80 members of this large AKH neuropeptide family are structurally known [6,11,12,13].

The first AKH that was structurally completely elucidated was from the two locust species *Locusta migratoria* (migratory locust) and *S. gregaria* (desert locust) and is a decapeptide (pGlu-Leu-Asn-Phe-Thr-Pro-Asn-Trp-Gly-Thr amide) which is today code-named Locmi-AKH-I [14]. About 10 years later, species-specific octapeptides were found in these two locust species: Locmi-AKH-II (pGlu-Leu-Asn-Phe-Ser-Ala-Gly-Trp amide) and Schgr-AKH-II (pGlu-Leu-Asn-Phe-Ser-Thr-Gly-Trp amide) [15,16]. In the migratory locust, a third octapeptide was isolated and sequenced, Locmi-AKH-III (pGlu-Leu-Asn-Phe-Thr-ProTrp-Trp amide) [17]. Another octapeptide was predicted [18] and found [19] in both locust species; this peptide, is called Aedae-AKH (pGlu-Leu-Thr-Phe-Thr-Pro-Ser-Trp amide) because it had been previously cloned and sequenced from the yellow fever mosquito, *Aedes aegypti* [20] and from the alderfly *Sialis lutaria* [11].

AKHs bind as peptides to GPCRs [21,22]. The AKH receptor (AKHR) for the desert locust has been cloned and is activated in vitro by the three endogenous AKHs using a calcium reporter assay [19,23]. The principle to use the insect peptide-GPCR system for the testing of agonists or antagonists is based on endogenous biologically active peptides, with the ultimate goal of synthesizing peptide mimetics, which interfere with insect physiological processes causing growth and retard development [24,25,26]. A good example is the use of beta blockers in the treatment of hypertension in humans. Beta-blockers bind to the β-adrenergic receptors of the sympathetic nervous system, thereby blocking the stress hormone, adrenaline, from binding. The design of such targeted GPCR treatment warranted the award of the Nobel Prize in Medicine to Sir J. Black in 1988.

To design “green” insecticides, we have previously characterized the *S. gregaria* AKH ligand–receptor system as follows [27]:The structures of the three endogenous AKHs were determined by nuclear magnetic resonance techniques in dodecylphosphocholine micelles; a turn structure was demonstrated for each peptide.A 3D model of Schgr-AKHR was constructed; the human kappa opioid receptor was found to be the best template.The AKHs were individually docked to the Schgr-AKHR, and a dynamic simulation of the ligand–receptor complexes in a model membrane was performed; the results demonstrated that the three AKHs bind to a common region on the receptor, interact with similar residues of the receptor, and have comparable binding constants.

In the current paper, this proposed workable model for the binding of the desert locust AKHs to its receptor is used in high throughput, virtual screening of a large number of possible antagonists, employing Schgr-AKH-II as our model ligand. Lead compounds were checked using extended precision screening and followed by molecular dynamics simulation in a phosphatidylcholine membrane. Finally, the antagonistic behavior of one of the promising compounds was checked in vivo with adult *S. gregaria*.

## 2. Materials and Methods

### 2.1. In Silico Screening

The previously determined structure of Schgr-AKH-II, (pELNFSTGW-amide) docked to Schgr-AKHR [27] was used as the initial docking construct for in silico screening. This GPCR was built using the 2.9 Å resolution crystal structure of hκ-OR (PDB ID: 4DJH), which was identified as the top template. In silico screening was performed using the virtual screening workflow (VSW) of the Schrödinger software package [28]. With this protocol, the ligands in the database were first prepared using the LigPrep [29] module of Maestro and the OPLS2005 force field [30]. Epik was used to apply the protonation states of the ligands at pH 7.0 [31,32]. The ligands were prefiltered with Lipinski’s rule [33], and then filtered to remove ligands with reactive functional groups. The active site on the Schgr-AKHR had been identified previously [27]. Using VSW, the prepared ligands were docked to the receptor active site using high throughput screening and the compounds ranked according to their glide scores [34]. The best 10% of poses were then subjected to standard precision docking [35], and the best 10% of poses from this stage were used in extended precision docking [36] (see Appendix A for workflow). Two databases were used for the in silico screening, the NCI Open 2012 library [37] (125723 compounds) and the Zinc20 database [38] from which the tranches CA, EA, FA, KA, KB and JA were extracted. For the Zinc20 tranches, compounds with logP < 0 and molecular weight 300–400 and >500 Daltons were selected. Only compounds that were commercially available and had no reactive groups were selected, resulting in a total of 94,000 protomers.

After extended precision screening, the compounds with the best glide scores and free energy of binding were selected for molecular dynamics in a 1-palmitoyl-2-oleoyl-sn-glycero-3-phosphocholine (POPC) membrane. Each ligand/receptor complex was placed in a POPC membrane and soaked with single point charge (SPC) water molecules using the Desmond System Builder [39]. The charge of the system was neutralized by adding chloride ions. The system was first relaxed in a series of stages over 1.2 ns using Desmond [39,40]. The relaxed system was subjected to 50 ns canonical ensemble (NVT) molecular dynamics simulation at 300 K. The results were analyzed using the Simulation Interaction Analysis task of Maestro [41]. Free energies of binding were calculated using the python script thermal_mmgbsa.py available from Schrödinger. All figures were prepared using Maestro.

### 2.2. Insects

Adult desert locusts, *Schistocerca gregaria*, were purchased from Fressnapf—a retailer store in Osnabrück, Germany. The sexes were kept apart and reared on fresh grass, dandelion leaves and oats supplied daily in large wire mesh cages (45 cm × 60 cm × 45 cm; L × H × W). Fecal matter and left-over food were removed from the cage daily. The locusts were kept at 23 °C, 40% RH, and a photoperiod of 17 h light: 7 h dark. The locusts were used in biological assays in the holding room. This study was carried out in accordance with relevant institutional and national guidelines and regulations concerning the use of animal subjects in scientific studies.

### 2.3. Biological Assay

A conspecific bioassay was carried out as described previously [42]. Briefly, male and female, adult *S. gregaria* were used in biological assays in the holding room. Each locust was put under a darkened funnel and left to rest for 1 h before sampling: 1 μL of hemolymph was collected with a disposable glass microcapillary (Hirschmann Laborgeräte) from a small wound made with the tip of a fine insect needle ventrally under the hind leg; the hemolymph was transferred into a test tube containing concentrated sulfuric acid; The locust was then injected ventrolaterally into the abdomen with 10 μL of an aqueous 1% dimethylsulfoxide (DMSO, Sigma-Aldrich, St. Louis, MO, USA) solution, or 10 μL of the synthetic peptide/compound under investigation (reconstituted in 1% DMSO in water). The locust was kept in the dark until a second sample of hemolymph was taken 90 min after injection.

In a second set of bioassays, a hemolymph sample was taken from resting locusts, which were then injected with ZINC257251537 (1466 pmol) and left to rest for 5 min before being injected, for a second time, with 10 pmol Schgr-AKH-II. After resting for 85 min a second hemolymph sample was withdrawn.

The hemolymph samples were thoroughly mixed with sulfuric acid, and the total vanillin-positive material (=lipids) measured in the mixture as described previously [42,43]. The difference in lipid concentration before and after injection was calculated for each individual animal and a paired *t*-test used for calculating statistical significance in Excel. An unequal variance *t*-test was used to test for significant differences between different groups of injected locusts.

### 2.4. Synthetic Peptide and Test Compound

The adipokinetic hormone II of the desert locust (Schgr-AKH-II) was synthesized originally by Peninsula Laboratories (Belmont, CA, USA). This endogenous peptide of the desert locust is known to have biological activity in an in vitro adipokinetic bioassay. The Zinc compound identified in our virtual screening of the Zinc20 database as a likely antagonist of Schgr-AKH-II, ZINC257251537 (molecular formula: C14H19N3O7, MW 341.317), was purchased as BDH33910066 from Asinex Corporation (Winston-Salem, NC, USA). Schgr-AKH-II was reconstituted in water and 10 μL (10 pmol) injected into the locusts. The Zinc compound was reconstituted in DMSO and injected into locusts at 500 pmol or 1466 pmol in 1% DMSO in a volume of 10 μL.

## 3. Results and Discussion

*Schistocerca gregaria* is one of the most destructive pest insects known to mankind. Devastating swarms of these locusts have plagued mankind for centuries, and the problem seems to have increased with global warming. The primary method of locust swarm control is the use of organophosphate pesticides applied in concentrated doses. Unfortunately, organophosphates are also highly toxic to humans. Since G-protein coupled receptors are a common target for drug development, and a unique GPCR is involved in the regulation of energy metabolism of locusts during flight, we turned our attention to the adipokinetic hormone signaling system of the desert locust to see whether a unique interference could be identified to target the pest signaling system and thereby, specifically reduce the mobility of locust swarms. The desert locust produces three adipokinetic hormones that bind to the same GPCR, Schgr-AKHR. In the current study we have focused on one of these AKHs, Schgr-AKH-II, with the objective to identify chemical antagonists of the Schgr-AKH-II neuropeptide, specifically, a competitive antagonist.

An antagonist blocks the effect of an agonist. There are two types of antagonists pursued by the pharmacology industry, viz. competitive (reversible, surmountable) and non-competitive (irreversible, insurmountable) antagonists [44]. Competitive antagonists bind to the same receptor site as the agonist, and their action can therefore be reversed when supplying the agonist in a sufficiently high dose, whereas non-competitive antagonists bind a different site on the receptor than the agonist. The most efficient way of screening a large number of compounds for receptor binding is to use in silico screening. Previously, we characterized the binding site of Schgr-AKH-II on the receptor, Schgr-AKHR, thus providing us with the ideal opportunity to screen several databases of chemical compounds. Since Schgr-AKH-II has a molecular weight of 885 and a log partition coefficient (logP) < 0 [45], our search was restricted to compounds with these characteristics. The Zinc20 database is the most comprehensive database of commercially available chemical compounds. Because of the size of this database, it is divided into tranches based on molecular weight and logP. Tranches of this database were downloaded and screened for receptor binding. High throughput screening of the CA tranche of 6332 compounds (MW < 300 and logP < −1) gave 277 potential lead compounds. For high throughput screening a glide score < −10 is considered good. The glide scores of the 277 compounds from the CA tranche were between −8.0 and −9.8. The free energy of binding of the compound with the best glide score (ZINC1574497495) was only −35.3 kcal mol^−1^. This is far less than the free energy of binding of Schgr-AKH-II (94–104 kcal mol^−1^) [27] and so it is unlikely that this compound would act as an antagonist and was rejected from further study. It is possible that the absence of a clear AKHR antagonist amongst the compounds in the CA tranche is linked to their MW of <300, whereas most common insecticides such as the organophosphate, Malathion, has a molecular weight of 330 and the pyrethroid insecticide, Bifenthrin, a MW of 422.

The subset aaml-cbrn of ZINC20 EA tranche (325 > MW < 350) contains 4500 compounds. After virtual screening, only one of these compounds was identified with a glide score < −10, ZINC257251537, 2-(2-(((1*S*,2*S*,3*S*)-2-hydroxy-3-(4-hydroxy-2-oxopyrimidin-1(2*H*)-yl)cyclohexyl)amino)-2-oxoethoxy)acetic acid, and after MD simulation, a free energy of binding of −60 ± 4 kcal mol^−1^ was calculated. A ligand–receptor interaction diagram (Figure 1) shows that the carboxyl group of the ZINC257251537 compound bonds strongly to Arg107 of the Schgr-AKHR via H-bonds, and with Cys269 via a water bridge. There is a water bridge between the carbonyls of the dioxo-pyrimidine and Ser88 for 30% of the simulation and the disulfide of the receptor for 48% of the simulation (Figure 1a). Lys281 forms water bridges with both the oxo group of the pyrimidine and the hydroxyl of the cyclohexane, and Trp93 has a π-π interaction with the pyrimidine for 70% of the simulation (Figure 1a). The protein–ligand contact diagram shows that there are other minor interactions between the ZINC257251537 compound and Schgr-AKHR but that these three receptor residues, Trp93, Arg107 and Lys281 are the most important (Figure 1b). All of these interactions lead to a free energy of binding of −60 kcal mol^−1^, and while this is less than most active AKH peptides, it is still a respectable binding energy.

Previous studies have shown [46] that amidation of the terminal carboxyl of an AKH is necessary for biological activity. For this reason, the acetate of ZINC25725137 was amidated and the MD simulations repeated. These results are shown in Figure 2a,b. After amidation there was no significant change in the free energy of binding (ΔG_bind_ = −60.8 ± 8 kcal mol^−1^) even though the H-bonding of the free acetate to Arg107 and Cys269 was lost. There was still a π-π interaction with Trp93, but now there was more extensive interactions with Pro82, Ser88, Ile85, Ser92 and Ala95 (Figure 2a). The docking of free and amidated ZINC257251537 can be compared with the docking of Schgr-AKH-II to the same receptor. For consistency, the MD of Schgr-AKH-II was repeated using the same protocol as for ZINC257251537 and then analyzed using the simulation interaction analysis. The results are given in Figure 3. ZINC257251537 is smaller than Schgr-AKH-II and so Schgr-AKH-II has more contacts with the receptor. The central section of Schgr-AKH-II (Asn3—Phe4) has similar interactions as ZINC257251537 with Trp93 and Arg107 (Figure 2a and Figure 3a). The C-terminus of Schgr-AKH-II fits into a pocket created by Pro82, Leu83, Ser92, Trp93, Arg94, Ala95 and Gly96. This is the same pocket of the receptor in which the pyrimidine of amidated ZINC257251537 is found (Figure 2a and Figure 3a). An overlay of amidated ZINC257251537 and Schgr-AKH-II in this Schgr-AKHR pocket is shown in Figure 2c. Given the −60 kcal mol^−1^ free energy of binding and the similarity of the Schgr-AKH-II and ZINC257251537 binding pockets, ZINC257251537 is a potential lead candidate as antagonist of Schgr-AKH-II.

Tranche FA (300 > MW < 325) gave 60 structures after extra-precision (XP) docking but only one, ZINC969533963 (Figure 4), had a glide score < −10. The root mean square fluctuations (RMSF) (Figure 4a) of the receptor are small, indicating that the ligand does not affect the stability of the receptor. The green vertical lines show that ZINC969533963 interacts with the second extracellular loops, ECL2, ECL4 and ECL6 of Schgr-AKHR. Figure 4b depicts the protein–ligand contacts normalized over the whole trajectory, broken down into categories: Trp174, Tyr175, Arg176 and Phe284, which are non-essential residues for the binding of Schgr-AKH, are the only receptor residues that make significant contact with the ligand; all of these residues make multiple contacts of different types such as: H-bonding, hydrophobic interaction and water bridge interactions. Figure 4c shows the ligand atoms that interact with the receptor. Here only one interaction with Tyr175 is maintained for 30% of the simulation. Figure 4b,c, taken together, indicate that the ligand is mobile within the binding pocket, and different ligand atoms interact with the receptor at different times. This lack of effective contact between ZINC969533963 and Schgr-AKHR is reflected in the free energy of binding which is only −32 kcal mol^−1^. This, together with the evidence that the zinc ligand binds to different receptor residues than seen in interactions between Schgr-AKH-II and Schgr-AKHR, makes ZINC969533963 an unsuitable competitive antagonist candidate of Schgr-AKH-II (see [44]).

Screening of the tranche JA (350 < MW > 375) gave only one compound with a glide score < −10. This compound, ZINC35189340, is a triphosphate with a very promising glide score of −12.83. MD of this compound revealed that the ligand contacts with the receptor were mainly via the phosphate groups, interacting with Trp93 and Arg107 in the binding pocket (not shown). Although these are two essential residues in the binding pocket of Schgr-AKHR, the free energy of binding was only −43 ± 5 kcal mol^−1^. The reason for this poor free energy of binding is poor van der Waals interaction (ΔG_vdW_ = −32(2) kcal mol^−1^), the generalized Born solvation energy (ΔG_SolvGB_ 12(3) kcal mol^−1^) and the lipophilic energy (ΔG_Lipo_ −10(1) kcal mol^−1^). Because of the low ΔG_bind_, and because the triphosphate is readily hydrolyzed in vivo, we simulated a modification of this compound by replacing the triphosphate with an alcohol group. Such a modified molecule gave a glide score of −8.9 and the ΔG_bind_ −27 ± 6 kcal mol^−1^; this free energy of binding was deemed insufficient to inhibit binding of the endogenous ligand.

Increasing the MW further and increasing to partition coefficient to include logP < 0, the KA (MW > 500, logP < −1) and KB (MW > 500, logP < 0) tranches of the ZINC20 database were screened. These tranches gave four compounds with good glide scores ranging from −10.0 to −11.1. All four compounds were peptidic. Gäde [46] has shown that peptides with unprotected termini are not biologically active; one reason for this may be that they are possibly hydrolyzed in circulation before reaching the receptor binding site and activating the AKHR on the fat body in insects. For this reason, and since we are looking for peptide mimetics, these compounds were not considered further.

Given the failure of the ZINC20 database to generate many promising lead compounds, the NCI database was used. This is a database of active compounds registered with the National Cancer Institute. Using this database of 125723 compounds, in silico screening gave 11 compounds with glide scores lower than −10. These compounds are listed in Table 1, together with their binding free energies. Compound 234446, which is a tribenzoate of dihydroxypentane (2,5-dihydroxypentane-1,3,4-triyl benzoate), had the best glide score of −10.9. On the other hand, compound 707401, a sulfonamide, ((*E*)-*N*-(amino((4-chlorophenyl)amino)methylene)-5-(phenylsulfonyl)thiophene-2-sulfonamide) and compound 211277, a derivative of pyrimidine (5-phenethyl-6-(3-phenoxypropyl)pyrimidine-2,4(1*H*,3*H*)-dione), had the best free energy of binding (−72.1 and −72.3 kcal mol^−1^, respectively). These three compounds were, therefore, chosen for more in-depth MD simulation in a POPC membrane. From the trajectory, the RMSF of the protein backbone was calculated as a function of time (Figure 5 for compound 707401 and Appendix A for other compounds). The residues of highest variability correspond to the loop regions of the receptor, while the areas of lowest flexibility correspond to the trans-membrane helices. The receptor with compound 211277 bound shows the most movement with ~4 Å fluctuation of extra cellular loop (ECL4) and intracellular loop (ICL5).

Movement of the seven trans-membrane helices (H1–H7) are <1 Å indicating that the receptor is not affected by the ligand and remains stable. The green vertical bars (Figure 5a) indicate residues that are in contact with the ligand: all three ligands interact with the same regions of the receptor indicating that they occupy the same binding site. Compound 707401 bound to Schgr-AKHR is shown in Figure 5b. Note that the ligand is bent, which is similar to the β-turn, typical of AKH peptides. The binding pocket comprises residues 55–80 of ECL2 and helix 2, residues 97–108 of helix 3, residues 187–192 of ECL4, and residues 269–272 of helix 6. This is the same binding pocket that was found for Schgr-AKH-II bound to Schgr-AKHR [27].

A 50 ns MD simulation in a POPC membrane was used to calculate the free energy of binding (Table 2) of the different ligands to Schgr-AKHR. The endogenous peptide, Schgr-AKH-II had the highest free energy of binding at −93 kcal mol^−1^. This agrees well with a previous 1 μs simulation, where the free energy of binding ranged from −94 to −116 kcal/mol [27]. Analysis of Table 2 shows that the major contributors to the free energy of binding are coulombic (−30 kcal mol^−1^), lipophilic (−31 kcal mol^−1^) and van der Waals (−88 kcal mol^−1^) interactions. Surprisingly, H-bonding contributes very little to the free energy of binding. On the other hand, the loss of entropy of the solvated ligand destabilizes the complex (ΔG_SolvGB_ = 58 kcal mol^−1^). The three ligands, 707401, 234446 and 211277, all have significantly lower free energies of binding than Schgr-AKH-II (two tailed *p*-values of 0.0001, 0.0305 and 0.0001, respectively). The main reason for this is the increased coulombic and van der Waals interaction of Schgr-AKH-II. Ligand 707401 has the lowest binding free energy because it does not have a favorable coulombic interaction with the receptor. On the positive side, all three ligands have only small entropic penalties upon binding. Note that water plays an important role in receptor binding. Firstly, there is the loss of water of solvation from the ligand, which has an enthalpy penalty. Then there is water in the binding site which helps the ligand binding by bridging between the ligand and receptor. Venkatakrishnan et al. [47] have emphasized the importance of water in the binding site. The more hydrophilic the ligand the better the binding to the receptor but the bigger the enthalpy penalty upon binding.

The ligand RMSF (Figure 6) is useful in monitoring the positional changes of the ligand within the receptor binding pocket. Of the three ligands, 234446 shows the greatest stability within the binding pocket, fluctuating < 1 Å. Ligand 707401 fluctuates < 2 Å, while the phenoxy group of ligand 211277 fluctuates ~ 4 Å. This end of the ligand moves within the binding pocket and does not interact with the receptor. The ligand interaction diagrams (Figure 7) shows that the phenoxy group of ligand 211277 only interacts with the receptor for 30% of the trajectory, π-π stacking with Tyr175. The benzyl end of the molecule π-π stacks with Trp272, 59% of the time. The most persistent interactions are the H-bonding of the pyrimidinedione with Cys178, Gln177 and His169. Ligand 234446 is an oxygen rich compound but only the one alcohol H-bonds to Cys178. The rest of the molecule sits in a hydrophobic pocket. Ligand 707401 has two sulfonamides, which H-bond to Arg107 and Ser92. There is also a water bridge with Cys178. An alternative way of displaying the data is given in Figure 8, where the fraction of time during the trajectory that the receptor interacts with the ligand is plotted against the receptor residues. Since any residue may interact with more than one ligand atom, a fraction greater than 1 is possible. Included in Figure 8 are the results for Schgr-AKH-II. Comparison of these diagrams for the different ligands shows that they all interact with Trp93 and Cys178. Comparing Schgr-AKH-II with the other ligands, the interactions are similar, but Schgr-AKH-II interacts for a larger percentage of the simulation time. One significant difference is that Schgr-AKH-II interacts with residues 93–96 on ECL2, while the other ligands only interact with Trp93. Ligand 211277 also interacts with a series of residues, His169–Cys178, which are on ECL4.

Our screening results can be compared with the in silico screening of a library of insecticide-like compounds (OTAVA Ltd., Vaughan, ON, Canada) to identify potential antagonists for the AKHR of the stick insect, *Carausius morosus* [48]. These authors found 12 candidate antagonists with ΔG_binding_ ranging from −33 to −93 kcal mol^−1^. The best free energies of binding of the 12 candidates identified are comparable to that of Schgr-AKH-II and the three NCI database ligands found in the current study (Table 2) but are better than the −60 kcal mol^−1^ found for ZINC25725137. The OTAVA Library compounds also have a similar structure to the compounds found here in that they all have aromatic groups and multiple H-bonding sites. The residue Arg269^3.32^ was critical for ligands binding to the stick insect AKHR [48]. This corresponds to Arg107^3.32^ of Schgr-AKHR, which is involved in H-bonding to Schgr-AKH-II, ZINC25725137 and ligands 707401 and 234446 (water bridge), but not 211277 (current study). Iyison et al. [48] found very stable H-bonding to Gln445^7.35^, an interaction with Tyr423^6.51^ and π-π stacking to Phe273^3.36^ and Trp431^6.59^ when characterizing ligand–receptor interactions of the stick insect AKHR. Our current data show that Schgr-AKH-II π-stacked to the equivalent residue Trp273^6.59^ of Schgr-AKHR, but this is not the case with the antagonist candidates; instead, compounds 211277 and 234446, π-stacked with the neighboring residue Trp272^6.58^, while neither Schgr-AKH-II nor any of our candidate antagonists interacted with helix7 (Gln445^7.35^). Schgr-AKH-II and all our compounds π-stacked with Trp93^3.18^ in the current study, whereas this residue was not found to be important for the stick insect [48].

Blind screening of a large number of compounds did not yield many lead compounds in the current study; in fact, there were no low molecular weight compounds that bound effectively to the receptor. Of all the ligands tried, compound 211277 had the best binding constant and the most interactions with the desert locust AKHR. For a compound to be an antagonist it must have a high affinity for the receptor but no efficacy (i.e., not produce a biological response). A biological/cellular response results from activation of a GPCR after binding of a ligand: a conformational change of the activated GPCR effects interaction of the intracellular part of the receptor with a heterotrimeric G-protein in the cytoplasm of the cell; the activated Gα subunit of the G-protein then interacts with a cytoplasmic effector molecule to produce a second messenger, which then brings about a cellular response in the classic signal transduction cascade [44]. In order to see if the ligand binding was sufficient to activate Schgr-AKHR in the current study, the distance between Leu248^6.34^ and Arg125^3.50^ was monitored. These residues are located on helices 3 and 6 in the intracellular region of the receptor, and it is known that upon activation of the receptor, helices 3 and 6 move apart opening a space for the G-protein to couple [49]. Figure 9 shows the distance between Leu248 and Arg125 over the course of the 50 ns simulation. The distance for all the compounds fluctuate ~1 Å with the mean distance of 9.8, 10.2 and 10.7 Å for compounds 211277, 707401 and 234446, respectively. Miao and McCammon [49] showed for the M_2_ muscarinic GPCR, in the inactive state the distance between Arg121^3.50^−Thr386^6.34^ is ~6–7 Å. When activated by the agonist, iperoxo, the distance increases to 14.5 Å. However, they also found two intermediate states where the distance was 10.0 Å and 12.0 Å. For the AKHR of the stick insect, *C. morosus,* it became clear that the TM3-TM6 distance was larger at 15.42 Å in the active state, but only 7.47 in the inactive state [48]. It is, therefore, worth speculating that the distances found for compounds 211277, 707401 and 234446 in the present study, represent the receptor in an intermediate state, not totally deactivated but also not active. Thus, these three ligands are possible antagonists and are worthy of further investigation.

Unfortunately, these three compounds are not readily available commercially. ZINC25725137 is, however, commercially available, and although this compound does not bind Schgr-AKHR as well as the three NCI compounds, we, nevertheless, tested its agonist and antagonist activity in a biological assay with *S. gregaria* in which the organism’s ability to increase circulating lipid levels under resting conditions is assessed (Table 3). Since the ZINC25725137 compound was reconstituted in DMSO, locusts were injected with 1% DMSO solution to ascertain a possible effect of the solvent; this had no significant effect on lipid concentration in the hemolymph. Similarly, ZINC25725137 in the micromolar concentration range did not mobilize lipid; an unequal variance T-test revealed that there was no significant difference between the effect of DMSO and the ZINC compound. This result is in accord with our simulation results which showed that ZINC25725137 did not activate the receptor (see Figure 9). On the other hand, 10 pmol of the endogenous AKH, Schgr-AKH-II or a *S. gregaria* corpus cardiacum extract that served as positive control, significantly increased the hemolymph lipid levels (Table 3). From these results it is clear that ZINC25725137 does not elicit a functional effect in vivo, but there is no evidence that the observed result is due to the compound not binding to the receptor or binding with no efficacy. Therefore, to determine if ZINC25725137 is, indeed, a competitive antagonist of Schgr-AKH-II, the in vivo bioassay was carried out with subsequent injections of the Zinc compound and the endogenous AKH—5 min apart—to see whether the agonist (AKH) could dislodge/compete with the putative antagonist (ZINC25725137) from/for the AKHR binding site. The results (Table 3) show that ZINC25725137 was able to suppress Schr-AKH-II induced lipid levels by 50% (*p* = 0.028). This is in accord with the simulation results (see Figure 2), which showed that ZINC25725137 bound to the same active site of the receptor as Schgr-AKH-II. Thus, ZINC25725137 does act as a competitive antagonist for one of the endogenous AKHs of the desert locust. The fact that Schgr-AKH-II could reverse the effect of a 146-fold more concentrated competitor to restore 50% of biological activity confirms our conclusion that ZINC25725137 is not an optimal competitive antagonist for blocking the locust AKH receptor. In a practical sense, this means that even higher doses of this chemical antagonist would be required to achieve an almost insurmountable block that cannot be lifted by endogenous hormone activity. The potential cost of application in such a case would very likely be unfavorably high compared to the current cost of organophosphates. Given that every 1 kcal increase in the free energy of binding leads to a 5.4-fold (at 25 °C) increase in the equilibrium binding constant, with better free energy of binding, we expect compounds 211277, 707401 and 234446, to have greater (optimal) binding to the AKH receptor and be more efficient competitive antagonists to Schgr-AKH-II. Thus, a much higher concentration of the natural hormone would be needed to reverse the effect of the putative insecticides in vivo.

Before any of these potential antagonists are selected as lead compounds for insecticide development, however, caution must be exercised to first ascertain that they meet the requirements of a “green insecticide”, i.e., they are not toxic to other animals (including non-target insects and humans), do not contaminate the food chain and do not damage to the environment (immediate physical effects and no long-term build-up).

## 4. Conclusions

In conclusion, in silico methods were used to screen 219,723 protomers for binding to the desert locust AKH receptor, Schgr-AKHR. Of the compounds screened, none bound to the receptor as well as the native ligand, Schgr-AKH-II. Several compounds did, however, bind to Schgr-AKHR with reasonable glide scores (<−10) and free energies of binding (<−60 kcal mol^−1^). Following MD simulation in a POPC membrane, four compounds were identified as potential antagonists. The MD simulations showed that they all bound to the same region of the receptor as Schgr-AKH-II but with a slightly lower free energy of binding. None of them activated the receptor. Unfortunately, none of these compounds were commercially available and so a related compound, ZINC25725137, which was available, and which we predicted would not be as good an antagonist as our four identified compounds, was used in a functional bioassay with the desert locust. When injected in a 1% DMSO solution, ZINC25725137, did not have any adipokinetic activity in *S. gregaria*. However, it did show significant antagonist activity in a competitive experiment with Schgr-AKH-II. Although our simulation experiments indicate that this is not the best candidate for an AKH antagonist, we have nonetheless shown here proof of concept that this may be a good candidate to limit AKH effect, and this raise hopes that the other candidates identified in the current study via in silico screening and further docking and simulations, may be even stronger antagonists.

## Figures and Tables

**Figure 1 life-12-00387-f001:**
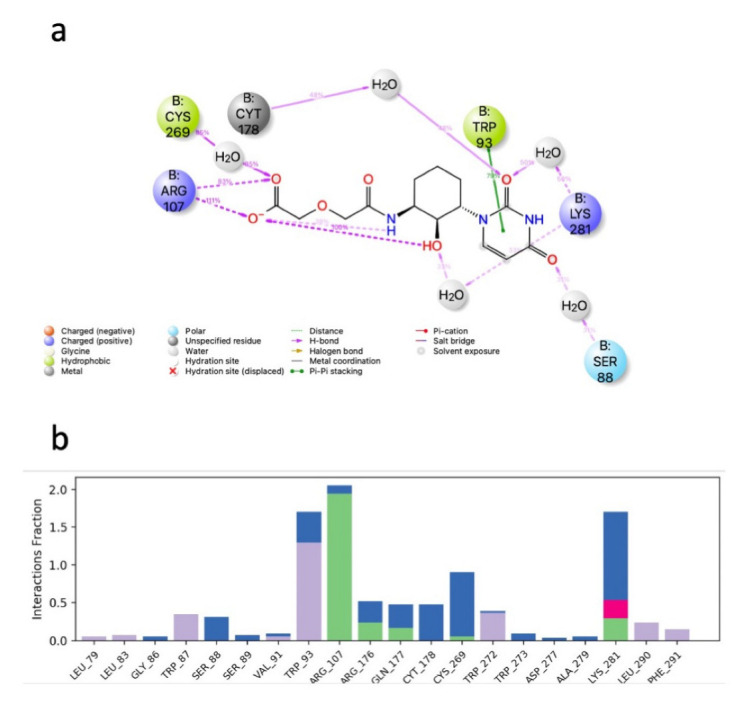
Interaction between Zinc25725137 and Schgr-AKHR. (**a**) Ligand interaction diagram. Lines show interactions that occur between the ligand and protein for more than 30% of the simulation. Purple = H-bonding and green = π-π stacking. (**b**) Protein ligand contacts. Green = H-bonds, blue = water bridges, grey = hydrophobic and red = ionic. Note Schrödinger uses the three-letter code CYT to denote cystine and CYS for cysteine.

**Figure 2 life-12-00387-f002:**
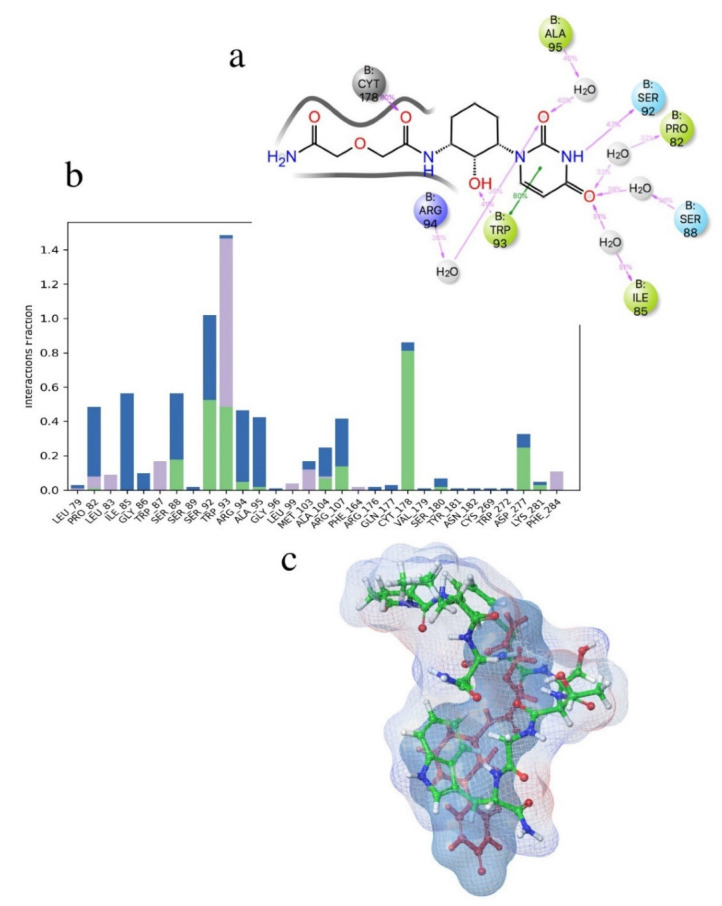
(**a**) Ligand interaction diagram of ZINC25725153amide. (**b**) Histogram of ZINC25725153amide interactions with Schgr-AKHR. Green = H-bond, blue = water bridge and grey = hydrophobic. (**c**) Overlay of Schgr-AKH-II and ZINC25725153amide in their binding pocket of Schgr-AKHR. ZINC25725153amide and its binding pocket are shown with a dark blue mesh. Schgr-AKH-II is shown with atom colors, and its binding pocket as an electrostatic potential mesh. Note Schrödinger uses the three-letter code CYT to denote cystine and CYS for cysteine.

**Figure 3 life-12-00387-f003:**
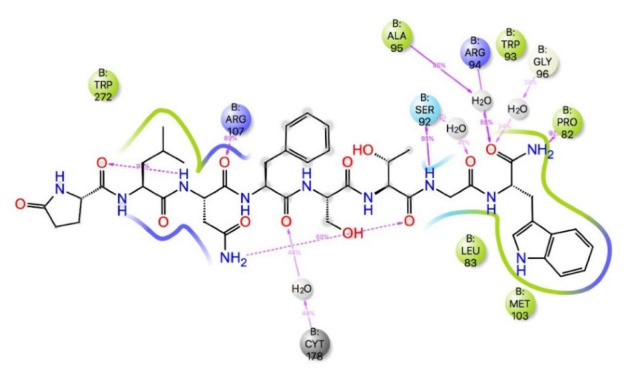
Ligand interaction diagram of Schgr-AKH-II and Schgr-AKHR. Note Schrödinger uses the three-letter code CYT to denote cystine and CYS for cysteine.

**Figure 4 life-12-00387-f004:**
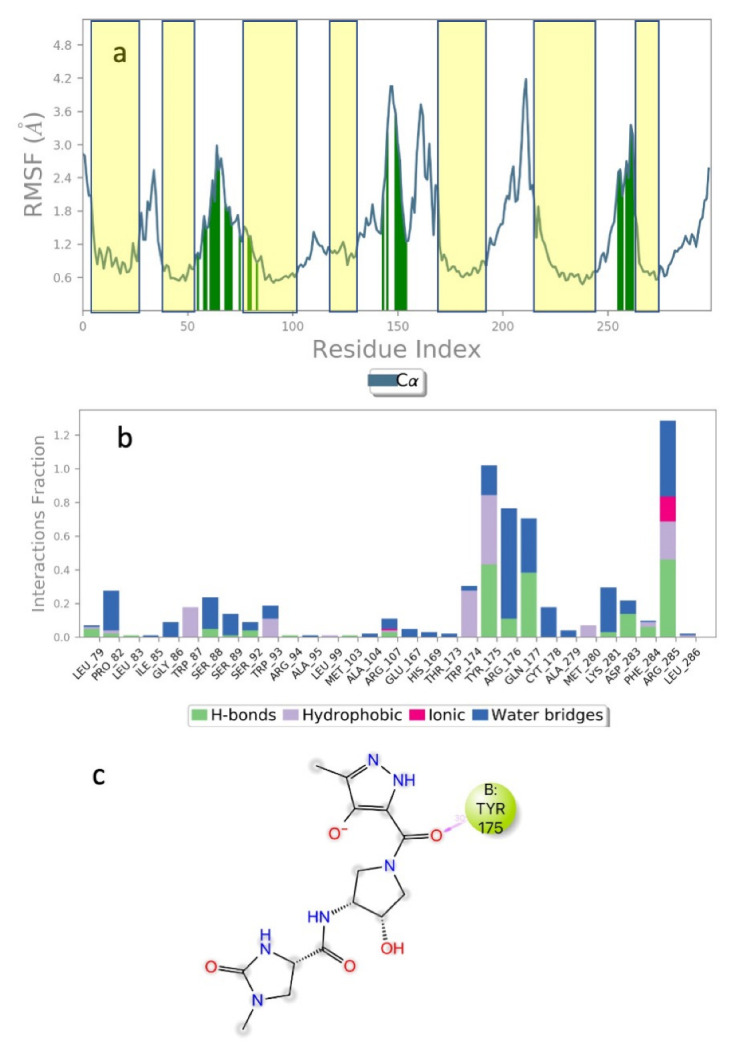
Interaction of ZINC969533963 with Schgr-AKHR during a 50 ns simulation in a POPC membrane. (**a**) Receptor RMSF of C_α_ during simulation. Vertical green lines show receptor residues that interact with ZINC969533963. Yellow shading indicates the position of the helices. (**b**) Histogram of ZINC969533963 interactions with receptor. Green = H-bond, blue = water bridge, grey = hydrophobic and red = ionic. (**c**) Ligand interaction diagram of ZINC969533963 to Schgr-AKHR. Note Schrödinger uses the three-letter code CYT to denote cystine and CYS for cysteine.

**Figure 5 life-12-00387-f005:**
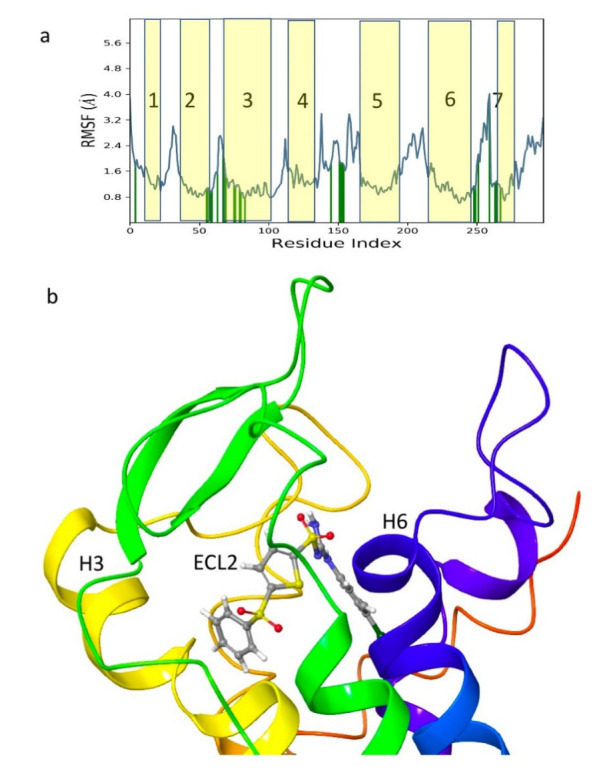
(**a**) Protein root mean square fluctuation (RMSF) of Schgr-AKHR C_α_ during a 50 ns MD simulation in a POPC membrane with compound 707401, from the NCI database, bound to the receptor. Vertical green lines show receptor residues that interact with the ligand. Yellow shading indicates the number transmembrane helices. (**b**) Compound 707401 in its Schgr-AKHR binding site.

**Figure 6 life-12-00387-f006:**
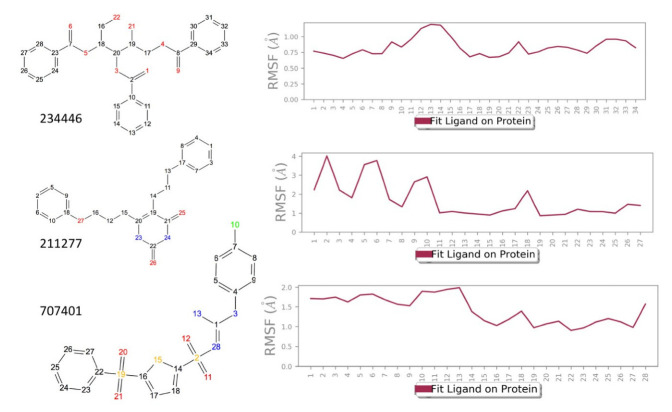
RMSF of ligand atoms while bound to Schgr-AKHR during a 50 ns simulation in a POPC membrane. Atom numbers are shown on the ligand structure.

**Figure 7 life-12-00387-f007:**
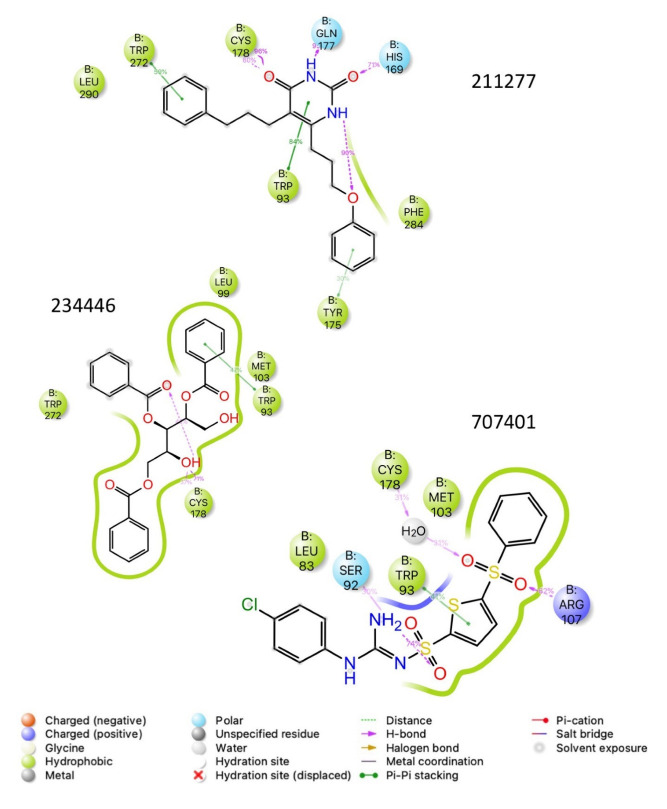
Schematic of protein–ligand contacts that occur for >30% of a 50 ns simulation in a POPC membrane.

**Figure 8 life-12-00387-f008:**
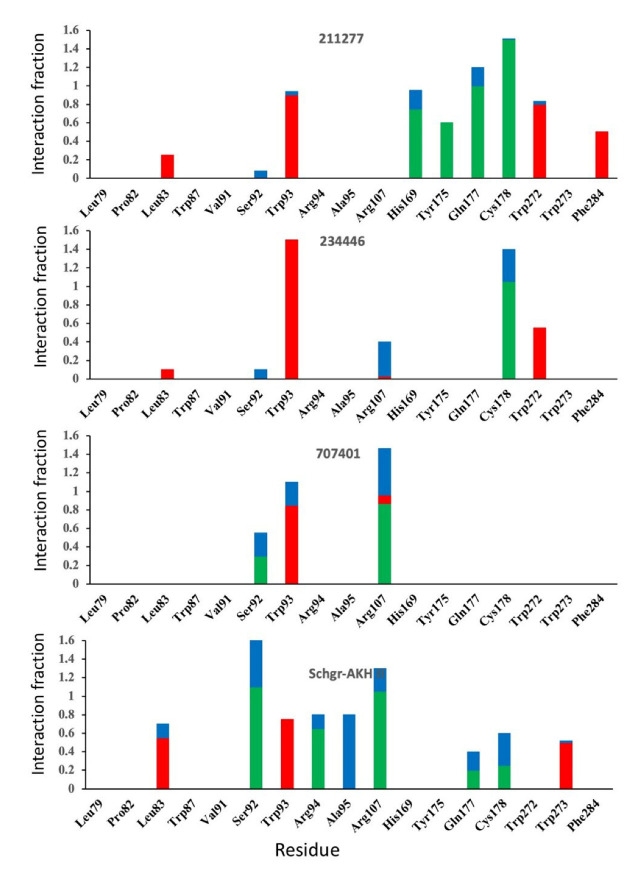
Histogram of protein–ligand contacts color coded by interaction type. Only contact that persist for more than 40% of the simulation are shown. Red = hydrophobic, blue = water bridge and green = H-bond.

**Figure 9 life-12-00387-f009:**
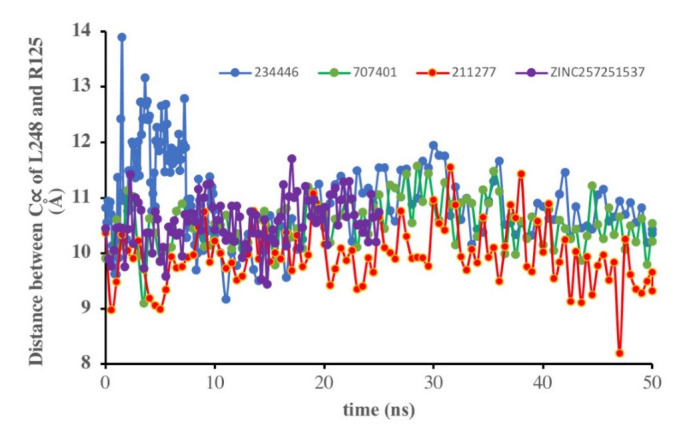
Plot of distance between C_α_ of Leu248 and Arg125 during 50 ns simulation in a POPC membrane with different ligands docked to Schgr-AKHR.

**Table 1 life-12-00387-t001:** Subset of compounds from NCI Database with a glide score (gScore) greater than −10 when docked to Schgr-AKHR in a virtual screening workflow.

Structure	Name	gScore	ΔG_bind_
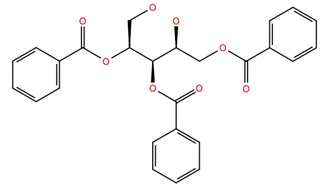	234446	−10.9	−60.1
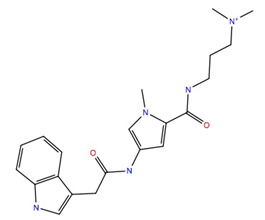	722562	−10.8	−52.5
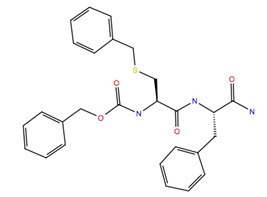	333451	−10.8	−58.5
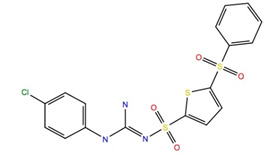	707401	−10.7	−72.1
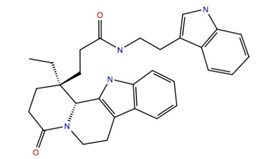	702438	−10.7	−36.2
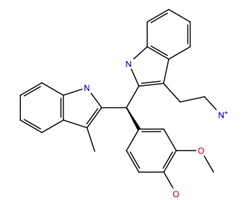	529379	−10.7	−45.4
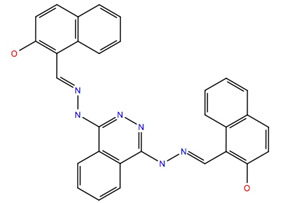	103663	−10.7	−60.7
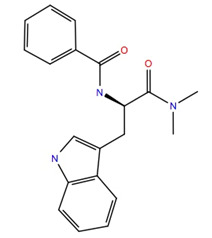	201298	−10.4	−60.3
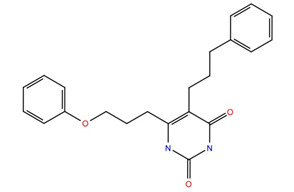	211277	−10.3	−72.3
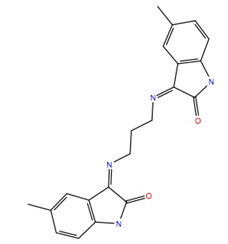	142427	−10.7	−53.9
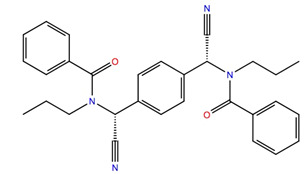	681749	−10.2	−48.0
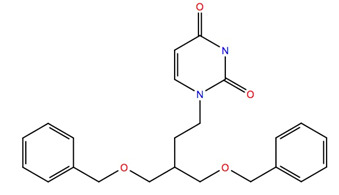	151836	−10.1	−67.8
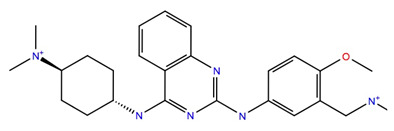	152001	−10.2	−56.3
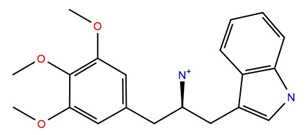	613572	−10.1	−53.1

**Table 2 life-12-00387-t002:** Breakdown of different energies (coulombic, covalent, H-bonding, lipophilic and GB solvation) giving rise to the free energy of binding of different ligands bound to Schgr-AKHR. Energies are averaged over a 50 ns simulation in a POPC membrane. Energies are in kcal mol^−1^ and standard deviations are given in brackets *.

Ligand	ΔG_bind_	ΔG_coulomb_	ΔG_covalent_	ΔG_H-bond_	ΔG_Lipophilic_	ΔG_SolvGB_	ΔG_vdW_
211277	−87(9)	−17(4)	0.7(1)	−1.3(0.3)	−29(3)	19(3)	−52(3)
234446	−80(4)	−16(2)	3.2(1)	−0.8(0.2)	−26(1)	22(2)	−61(3)
707401	−72(6)	4(5)	1.6(1)	−1(0.5)	−21(2)	7(4)	−56(4)
Schgr-AKH II	−93(10)	−30(9)	−0.3(3)	−1.3(1)	−31(3)	58(7)	−88(7)

* ΔG_vdW_ = van der Waals interaction, ΔG_SolvGB_ = generalized Born solvation energy, ΔG_coulomb_ = coulombic energy, ΔG_H-bond_ = H-bonding energy and ΔG_Lipo_ = lipophilic energy.

**Table 3 life-12-00387-t003:** Biological activity of a crude methanolic extract of corpora cardiaca from the desert locust (*Schistocerca gregaria*), and the synthetic peptide Schgr-AKH-II in homologous bioassays, as well as the activity of ZINC25725137 on its own or challenged with Schgr-AKH-II.

Treatment	*n*	[Lipid]T_0_min (µg/µL)	[Lipid]T_90_min (µg/µL)	Difference (µg/µL)	*p* *
1% DMSO	6	8.51 ± 2.04	7.36 ± 3.03	−1.14 ± 2.84	NS
ZINC25725137 (500 pmol)	5	8.87 ± 1.62	9.23 ± 2.15	0.35 ± 1.32	NS
ZINC25725137 (1466 pmol)	10	6.98 ± 1.15	7.74 ± 2.04	0.76 ± 1.48	NS
ZINC25725137 (1466 pmol) challenged after 5 min with Schgr-AKH-II (10 pmol)	15	7.79 ± 1.04	13.14 ± 2.63	5.35 ± 2.35	≤0.001
Schgr-AKH-II (10 pmol)	10	10.26 ± 4.45	20.69 ± 10.51	10.44 ± 7.23	≤0.001
*S. gregaria* corpora cardiaca (0.1 gland pair equivalent)	5	7.61 ± 1.35	21.62 ± 10.05	14.01 ± 8.93	≤0.001

Data given as mean ± SD. * A Paired *t*-test was applied to compare data before and after injection in the same individuals. NS = not significant.

## Data Availability

Not applicable.

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
