# Peer review of "In Silico Screening for Pesticide Candidates against the Desert Locust Schistocerca gregaria"

_life, 2022, doi:10.3390/life12030387_

Round 1

Reviewer 1 Report

This is a well designed approach to find analogues that block the activity of AKH in locusts that cause extensive and devastating damages throughout the world.  There were few correction to the text that I made on the body of the manuscript (see enclosed pdf).  I would like the authors to a.  Expand the discussion and explain based on the free energies of the different compounds why 211277, 707401 and 234446 with Gibbs free energies of -87 kcal/mol, -72 kcal/mol and -80 kcal/mol will be much better binders than the ZINC25725137 with Gibbs free energy of -60 kcal/mol.  I admit that the authors did explain this very briefly in the conclusions section, however, more in depth explanation to those biologists that are not familiar with physical chemistry is warranted.  I am questioning the p value of 0.0000002 in Table 3.  Is this a typo mistake?  The authors should also discuss in more details that using 140-fold more of the competitor than the natural hormone, and preventing only 50% binding is not the optimal approach to block AKHII from binding. Hopefully the other 3 compounds with more optimal binding will be more efficient.  I also would like to see that the authors explain that these organic compounds, albeit different than the organo phosphate analogues that are now used in pesticides will have to be tested to make sure that they are not toxic to other insects, animals , humans and do not cause damage to the environment.

Reviewer 2 Report

In order to identify compounds which may fit the endogenous Schgr-AKH-II binding site on the AKH receptor against the desert locust (Schistocerca gregaria), this manuscript presents: 1) in silico approaches, consisting in: screening of two databases, molecular docking, molecular dynamic simulation; and for one compound in vivo testing. Although the workload and computational methods used were somewhat laborious, unfortunately the end results were not great. The only tested compound (ZINC25725137) does not elicit a functional effect in vivo. However, the value of the theoretical study remains. As a suggestion for future studies, a different approach, not based on a selection of compounds in the first instance based on the docking score, could be more productive. It would probably be more successful to select the compounds taking into account the key interactions in the binding site. Can also be used pharmacophore or shape similarity methods.

A recommendation, a scheme of the workflow could be introduced.

Please correct the gScore of the compound "152001", Table 1. -10.2 instead of 10.2.
